# Financial Uncertainty from a Dual Shock at Global Level–Insights from Kuwait

**Talal A. N. M. S. Alotaibi and Lucía Morales \***

Department of Accounting, Economics and Finance, Technological University Dublin, D07 EWV4 Dublin, Ireland
\* Correspondence: lucia.morales@tudublin.ie

**Abstract:** Global stock markets experienced a dual shock in 2020 due to the impact of the global health crisis, parallel to a simultaneous shock derived from the Saudi Arabia and Russia oil price war. The dual shock fueled oil market volatility with lasting effects as the global economy is immersed in an energy crisis combined with high inflationary pressures exacerbated by heightened energy costs. This research paper implemented GARCH and FIGARCH models on daily returns from 31 December 2015, to 9 December 2021, to examine volatility persistence and long memory processes. The world's most prominent economies are represented by the G7, E7 and the GCC stock markets. Particular attention was devoted to the case of Kuwait as an example of a small oil-dependent economy. The research findings suggest evidence of volatility persistence across the markets, as reported by the GARCH (1,1) model. The FIGARCH (1,1) did not offer significant evidence of long memory processes except for the cases of FTSE 100, BIST 100, IDEX, BSE 100 and Bahrain.

**Keywords:** Kuwait; COVID-19; oil; market shocks; stock markets; dual shock

## 1. Introduction

In 2020 the world economies faced a dual shock due to the COVID-19 pandemic and the oil price war between the Kingdom of Saudi Arabia and Russia. As the global health crisis escalated, governments worldwide took an active approach to counteract the spread of the novel coronavirus. The introduction of social distancing measures led to economic lockdowns and economic hibernation with significant socio-economic implications (Morales and Andreosso-O'Callaghan 2020). The decision to enter into economic hibernation and the introduction of stringent social distancing measures led to social and economic hardships and acute disruption of global supply chains that resulted in a severe reduction of global aggregated demand. The situation aggravated in 2021 as inflationary pressures emerged, negatively affecting the world economies (Jackson et al. 2020; Ha et al. 2021). Discrepancies between Saudi Arabia and Russia led to significant oil supply disruptions that added additional pressures to oil-exporting economies. Furthermore, most oil-exporting economies are affected by a lack of economic diversification, a combination of increasing unsustainable levels of debt backed by oil collateral. The GCC region is significantly exposed to oil market dynamics, combined with the rise of illicit financial flows that threaten the countries' competitiveness, increase their risk exposure and threaten their social, political and economic stability (OECD 2020).

The 2020 dual economic shock caused significant disruption of global supply chains by reducing aggregated demand and disrupting international trade flows. Global health and oil market dynamics brought significant uncertainty levels to the world economies and had significant implications for oil-exporting economies like the case of Kuwait. On 9 March, the Kingdom of Saudi Arabia began an oil price war against Russia by increasing production levels by 25%, reaching 12.3 million barrels daily. The decision caused a significant disruption in the oil market, with oil prices plunging as oil barrels were sold at historically low prices, with an immediate 30% price decline (Jawadi and Sellami 2021).

According to Albulescu (2020), this shock led to a financial market crash on the same day, coined as Black Monday. The situation did not improve even after the historical deal of the OPEC countries agreeing to cut oil production by 10 million barrels/day on 12 April 2020, as prices were 22$ (WTI) and 18$ (Brent) on 13 April, and they moved into negative territory towards the end of the month (Ruiz Estrada 2020). Economic oil-dependent economies, as in the case of Kuwait, are characterised by a fragile economic system, which relies heavily on oil exports and lacks a well-diversified economy that could support a smooth transition towards cleaner and more sustainable business activities.

Undoubtedly, oil plays a critical role in Kuwait's economy as the country has a significant time-varying financial dependency on fossil fuels. Oil and natural gas account for nearly 60% of GDP and about 92% of export revenues. Within the outlined context, the purpose of this paper is to analyse the impact of COVID-19 and the oil shock on the return and volatility of the Kuwait stock exchange along with major markets indices of the top G7 (the world most developed economies), the E7 (the most relevant emerging economies) and the GCC (the Gulf Cooperation Countries) countries. The reviewed literature reveals that existing research studies have not integrated the world's major stock markets to frame the performance of the Kuwait stock exchange (Boursa) amidst a dual market shock named the global health crisis and the oil price war in 2020. Furthermore, this research study is supported by two GARCH specifications, namely GARCH and FIGARCH models seeking to examine volatility persistence and long memory processes. Volatility modelling is implemented to explore the performance of the Kuwaiti stock exchange among the world's major indices and to examine volatility patterns to identify which markets have shown more resilience dynamics to the 2020 dual shock. The results showed that the GARCH(1,1) helped to explain volatility persistence dynamics in the studied markets. However, the FIGARCH (1,1) did not offer significant evidence of long memory processes affecting the studied markets except for FTSE 100, BIST 100, IDEX, BSE 100, and Bahrain. Certainly, oil-rich economies face a significant dilemma as they explore the transition towards a more sustainable economic model amidst the evident global dependency on fossil fuels, as revealed by the Russian-Ukrainian war (European Commission 2022).

The rest of the paper proceeds as follows. Section 2 reviews the relevant literature; Section 3 defines the methodological research framework. The paper's main results and discussion are presented in Sections 4 and 5, and finally, Section 6 concludes the paper.

## 2. 2020 Dual Economic Shock

A brief historic insight reveals that the 20th century has witnessed three pandemics; the historical Spanish influenza in 1918, the Asian flu in 1957; and the Hong Kong flu in 1968. On the other hand, the 21st century has seen four pandemic outbreaks: the Severe Acute Respiratory Syndrome (SARS) in 2002; the Bird Flu in 2009; the Middle East Respiratory Syndrome (MERS) in 2012; and Ebola in 2013, clearly flagging that health crises are not new phenomena (Baldwin and Di Mauro 2020).

The most recent outbreak was first reported in Wuhan, China, in December 2019, and since then, it has spread worldwide. The first death case related to COVID-19 was reported on 11 January 2020, by the World Health Organization (WHO 2020)[1]. On 11 March 2020, the WHO declared COVID-19 as a global pandemic, and by 14 July 2022, the number of registered cases at the global level accounted for 556,897,312, and the total number of deaths was 6,356,812. In the case of Kuwait, the number of reported cases is around 648,216, and the total number of deaths is 2556 (WHO 2022).

In the economic and financial context, the emergence of COVID-19 has caused financial markets to suffer historic losses in the first quarter of 2020 at levels unseen since the crisis of the futures markets in 1987, followed by spillover effects to the macroeconomy (BBC, 31 March 2020). For instance, the Dow Jones Industrials, the S&P 500 and the NASDAQ (the technological index) declined 3.5%, 3.3% and 3.7%, respectively, during the initial stages of the COVID-19 pandemic and subsequently increasing levels of unemployment (BBC, 24 February 2020). As the global health crisis escalated, an oil crisis was also in the

making. On March 9, the Kingdom of Saudi Arabia began an oil price war against Russia by increasing production levels by 25%, reaching 12.3 million barrels a day. The decision caused a significant disruption in the oil market, with oil prices plunging as oil barrels were sold at historically low prices (Jawadi and Sellami 2021; Albulescu 2020). The situation did not improve even after the historical deal of the OPEC countries to cut oil production by 10 million barrels/day on 12 April 2020, as WTI prices were at 22$ and Brent prices at18$ on April 13. Figure 1 below illustrates how oil prices continued their sharp decline through April 2020, and by the 19th, the West Texas Intermediate Index (WTI) reached 17$. Moreover, the 20th and 21st of April were historical days as WTI recorded on the 20th negative prices of −36$ and the next day 21st −6$ for the first time in oil history (Alotaibi and Morales 2022; Ruiz Estrada 2020).

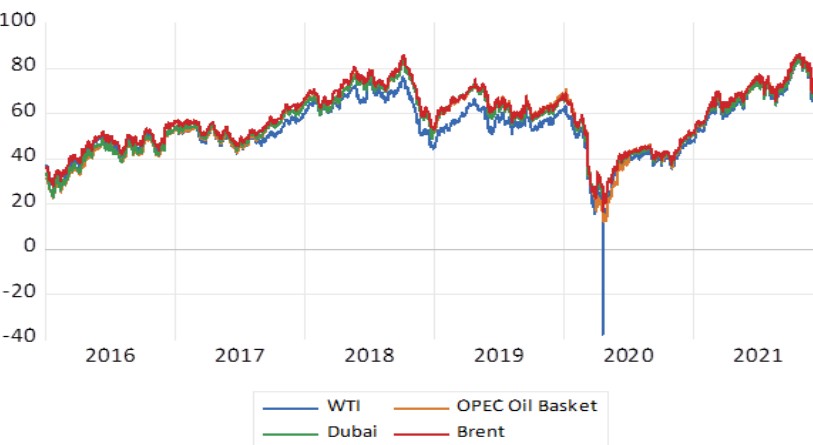

**Figure 1.** Oil Prices. Note: WTI recorded a historically low price of −36$ in April 2020. Source: Datastream 2022.

Kuwait is one of a few countries that faced a double shock in 2020, as the global health crisis was enhanced by significant levels of uncertainty emerging from the oil markets, leading to an unsustainable economic situation for the country. As an oil-exporting country, Kuwait's 2020 financial budget was significantly impacted, as oil prices were forecasted at $55 per barrel. GDP at constant prices declined by 8.9% compared to an average growth rate of 0.4% in 2019 (Jawadi and Sellami 2021). The effect of oil on stock markets is a vital area of discussion and analysis in financial economics, especially in the context of oil-producing countries such as Kuwait. Accordingly, researchers and practitioners have devoted significant attention to analyse the impact of oil fluctuation on the Kuwaiti's stock market (KSE) due to its important repercussions for the country's economic and financial system. Over the past decade, researchers such as Al-Shami and Ibrahim (2013); Al Hayky and Naim (2016); Merza and Almusawi (2016); Elian and Kisswani (2018); Al-Kandari and Abul (2019); Yousef (2020); Abdulrazzaq et al. (2019); Alshihab and Al Shammari (2020); Al Refai et al. (2022) have considered the impact of oil fluctuations on Kuwaiti stock market from two different perspectives, the macroeconomic and the financial dimensions. The research findings offer significant evidence of a positive relationship between oil price dynamics, the stock market's performance, and its spillover effects on the country's economy.

A summary of the extant literature and the core research outcomes focused on the case of Kuwait is presented in Table A1 (see Appendix A). Overall, the research findings highlight the lack of research in a global context and under the 2020 dual shock, being these aspects the focus of interest and the critical contribution of this research paper.

### 2.1. The Impact of COVID-19 on Global Stock Markets

The global stock market crash in 1987 represented a breakpoint for financial markets, as it became critical for fund managers and policymakers to understand how international

financial markets affect each other. Volatility spillover effects are defined as the transmission of instability from one market to another. When volatility prices change in one market, it causes a lagged impact on volatility prices in another market that is above the local market effect (Engle et al. 1990). In this context, the COVID-19 pandemic can be considered a good example of this phenomenon, as it affected the financial system's stability and public health care. Figure 2 below highlights the effects of the pandemic on the G7 markets, including one emerging market (Kuwait), to gain insights into the performance of the Kuwait stock market, as this market is the focal point of this study. The graph illustrates that 23 March 2020, recorded the lowest point across markets. The American index (Dow Jones Industrials), the Italian index (FTSE MIB) and the Canadian index (S&P TSX) were the most impacted, as they dropped by approximately 37%. This is followed by the German index (DAX 30) and the French index (CAC-40) as both indexes dropped by 36%, with the British index (FTSE100) and the American (S&P500) both recording a negative 34%, and the Japanese index (NIKKEI) registering a negative 29%. Kuwait differentiates itself from the G7 economies as being a small economy that is highly dependent on oil. Consequently, it was quite surprising that Kuwait performed better than the G7 economies. The BK All Share Index registered the lowest drop, with a 25% decline as illustrated in Figure 2 below (DataStream 2022).

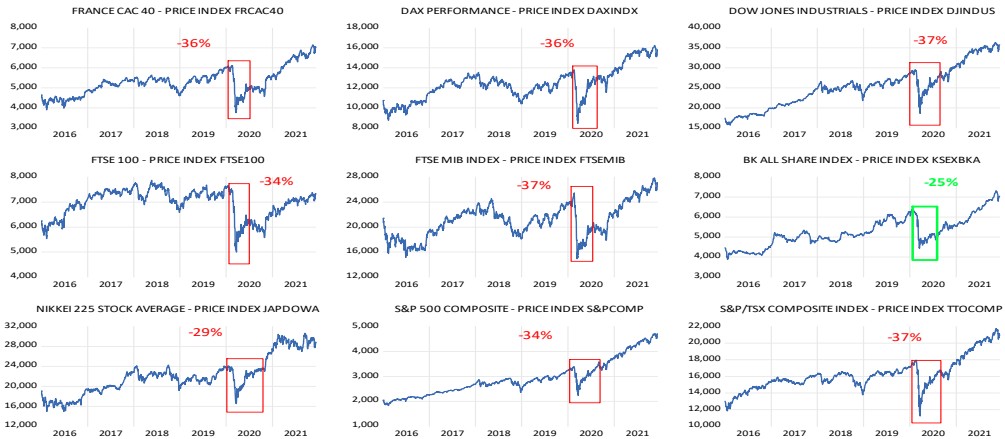

**Figure 2.** G7 Stock Markets Decline amidst the COVID-19 Pandemic. Note: The figure highlights the lowest points reached by the G7 leading stock markets, which were recorded on 23 March 2020. Source: DataStream (2021).

For comparative purposes, Figure 3 below highlights the volatility shock of the E7 markets, including the Kuwait stock market. The graph illustrates that 23 March 2020, was also the lowest point for the world's major emerging economies and in alignment with the performance of the G7 markets. The Brazilian index (BOVESPA) was the most impacted as it dropped by 46%, followed by the Indian index (S&P BSE100), which fell by 38%. The Russian index (RTS), Indonesian index (IDX) and Turkish index (BIST100) followed suit registering a 34%, 33% and 30% decline, respectively. On the other hand, the Mexican index (BOLSA) recorded the lowest impact with a negative 27%. The Chinese index (SHANGHAI) recorded the least significant fall, with a 9% drop that is justified by China's adjustment to the crisis, as the world stock markets were lagging and did not react to the worrying news emerging from China by the end of 2019. According to Morales and Andreosso-O'Callaghan (2020), the Shanghai index experienced a short-term impact on global markets during the early days of the outbreak as it was generally quite disconnected from worldwide panic tendencies. China's reaction to the virus was focused on stabilising and controlling the spread of the virus and reacted more strongly to its effects than the rest of the world economies. In the context of the E7 stock markets, the Kuwaiti index performed relatively better, except for the case of the Chinese stock market, represented by the Shanghai Composite Index (DataStream 2022).

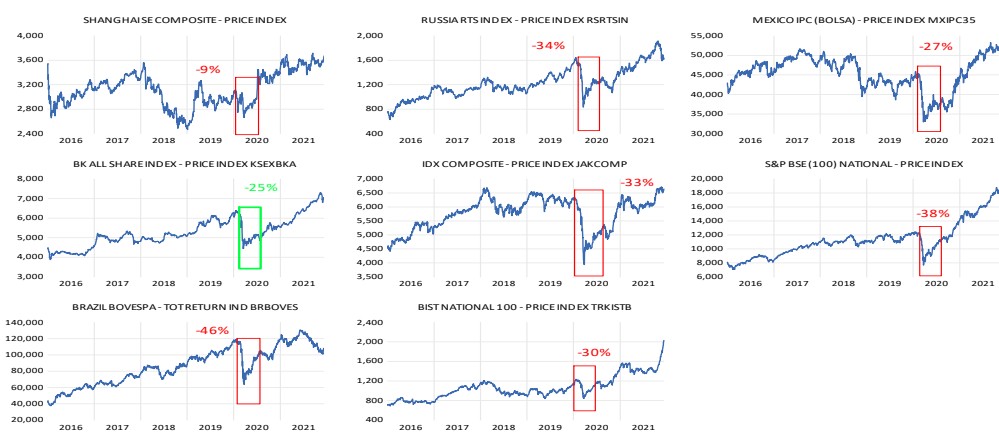

**Figure 3.** E7 Stock Markets Decline amidst the COVID-19 Pandemic. Note: The figure highlights the lowest points reached by the E7 leading stock markets recorded on 23 March 2020. Source: DataStream (2021).

Figure 4 below highlights the effects of the global health crisis on the performance of the GCC markets. Aligned with the outcomes for the G7 and the E7 markets, significant price drops were recorded on 23 March 2020. The Dubai index (DFMGI) was the most impacted as it dropped by 37%, followed by the Kuwaiti index (KSE), which dropped by 25%, followed by the Saudi index (TASI), which fell by 24%. On the other hand, the Bahrain index (Bahrain), Qatar index (Qatar) and Oman (Muscat) recorded the lowest impact with drops of 16%, 14% and 13%, respectively.

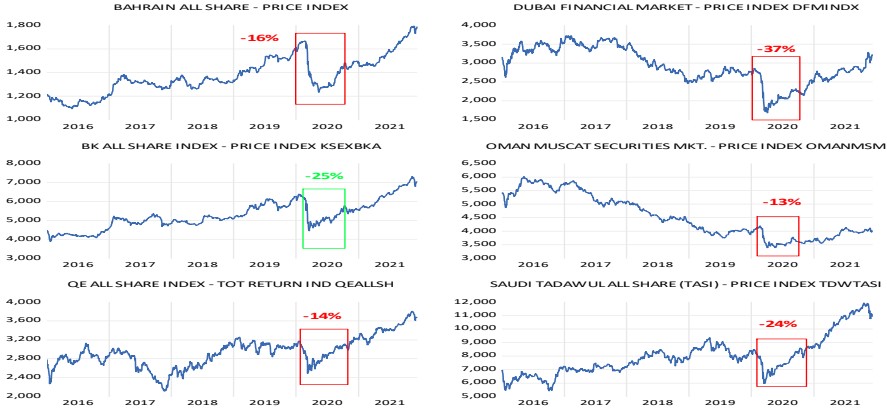

**Figure 4.** G.C.C. Stock Markets Decline amidst the COVID-19 Pandemic. Note: The figure highlights the lowest points reached by the GCC leading stock markets, which were recorded on 23 March 2020. Source: DataStream (2021).

## 2.2. Volatility Performance of the Kuwait Stock Market in a Global Context

Al-Kandari and Abul (2020); Al Ajmi (2020) used GARCH modelling to analyse the Kuwait stock exchange, as Al-Kandari and Abul (2020) studied the impact of the regulatory changes in the Kuwait stock exchange in terms of volatility. They applied ARCH, GARCH and TGARCH models to examine the market's volatility in two sub-periods. Their findings suggest that the Kuwait stock exchange was more volatile during the pre-liberalisation period compared with the liberalisation period. Hence, their study indicated that the T-GARCH is the best model for estimating the volatilities of Kuwait's stock exchange returns. In the same line, Al Ajmi (2020) investigated the conditional variances, in daily returns of Boursa Kuwait market index, along with seven sectoral indices, from 13 May 2012, to 1 March 2018, using three GARCH models (GARCH, EGARCH, and TGARCH). The GARCH-M model showed a negative relationship between the indices' returns and risk. The research findings inferred that good news has a more significant impact than bad

news on the volatility of index returns. Furthermore, Alotaibi and Morales (2022); Yousef (2020) examined the impact of the dual shock (global health crisis and oil price shock) in the GCC economies finding evidence of significant disruption across the region, except for the case of Bahrain's stock market that emerged as being relatively stable.

The reviewed literature shows that the most updated papers on the effects of COVID-19 have focused on analysing the world's most developed stock markets. Few studies have established a link between the global health crisis and the oil prices shock in a research framework that considers the impact of a dual shock in emerging markets such as Kuwait. Moreover, over the past ten years, most of the literature examining the case of the Kuwaiti stock market has been focused on studying the macroeconomic and microeconomic perspectives confirming the positive relationship between the Kuwait stock market and oil price fluctuations. Hence, the extant literature shows a dearth of research studies examining volatility dynamics in the context of GARCH models like the well-known GARCH and FIGARCH to examine the effects of the COVID-19 pandemic in the Kuwait stock market and the integration of the oil shock, an issue that we address in the next section.

## 3. Research Methodology and Methods

The data set comprises the leading indexes from the world stock markets represented by: Kuwait's weighted market index and the world's most relevant markets depicted by the G7, E7, and GCC stock markets (see Table A2 in the Appendix B for details). Additionally, the data set integrates four crude oil benchmarks represented by the US West Texas Intermediate (WTI), the European Brent Index (Brent), Dubai Crude oil (Dubai) and OPEC reference basket (OPEC). Oil benchmarks were chosen based on secondary data for continuous returns downloaded from DataStream over the historical period available for the Kuwait index between 31 December 2015, and 9 December 2021. Following the recommendation of Ng and Lam (2006), the research study sought to gather a minimum of 1000 observations to ensure that the GARCH modelling exercise did not encounter problems due to data limitations. Accordingly, a higher data frequency is needed to capture the changes in the market. In this study, daily closing prices were used, resulting in a total of 1551 observations. The data set providing details of all the variables included in this study can be found in Table A2 in the Appendix B.

### 3.1. GARCH and FIGARCH Models

### 3.1.1. GARCH (p,q)

The GARCH model extends the autoregressive conditional heteroscedasticity model (ARCH). The ARCH model presented by Engle (1982) suggests that the conditional variance equation needs to be exhibited as a linear function of the past periods (q) model represented in Equation (1) below:

$$h_t^2 = \omega + \sum_{i=1}^{q} \alpha_i \varepsilon_{t-i}^2 \tag{1}$$

where $\omega$ and $\alpha_i$ are non-negative parameters to ensure that the conditional variance is positive and $\varepsilon_{t-i}^2$ is the square error obtained from the mean equation. The fit of the ARCH (q) model for financial time series has worked well only when using a large number of lags. This weakness led to numerous extensions of this model. One of the most important contributions is Bollerslev's generalised autoregressive conditional heteroscedasticity (GARCH) model (Bollerslev 1986). The GARCH model attempts to overcome the need for a large number of lags to correct the model for the high persistence of variance associated with financial and economic data. The GARCH (p, q) model is different from the ARCH (q) model as it models the conditional variance as an autoregressive moving average ARMA process such that the innovations and their lags determine the conditional variance. To do this, the GARCH (p, q) model jointly estimates two equations, the conditional mean equation and the conditional variance equation, which for the (q) is the lag length of the autoregressive component and the (p) is the lag length of the moving average component. The two equations below support this study.

Mean equations

$$r_t = \mu + \varepsilon_t \tag{2}$$

or

$$r_t = \mu + \sum_{i=1}^{p} \alpha_i r_{t-i} + \sum_{j=1}^{q} \gamma_j \varepsilon_{t-j} + \varepsilon_t \tag{3}$$

where, $r_t$ represent the daily return of a market index, $r_{t-i}$ and $\varepsilon_{t-j}$ are the autoregressive and moving average components, respectively, and q and p are the lag orders of the processes.

This is referred to as the Conditional Variance equation since $h_t$ is the one-period-ahead variance forecast based on past information, called conditional variance. Hence, the conditional variance is the fundamental contribution of the GARCH (p, q) model, and can be written as represented in Equation (4):

$$\varepsilon_t \mid \Omega_{t-1} \sim N\left(0, h_t^2\right),$$
$$h_t^2 = \omega + \sum_{i=1}^{p} \alpha_i \varepsilon_{i-1}^2 + \sum_{j=1}^{q} \beta_j h_{t-j}^2 \tag{4}$$
$$\omega > 0, \alpha_i, \beta_j \geq 0 \rightarrow h_t^2 \geq 0, i = 1, \ldots p, \text{ and } j = 1, \ldots q$$

where $\Omega_{t-1}$ is the set of all information available at time $t-1$. The conditional variance of the GARCH model is defined in Equation (4) in three terms. The first term is the mean of yesterday's forecast, $\omega$. The second term is the lag of the squared residual taken from the mean equation, $\varepsilon_{i-1}^2$, or the ARCH terms. The ARCH terms represent news (information) about volatility from the previous period that has a weight impact, which declines gradually, never reaching zero, on the current conditional volatility. The third term is the GARCH term, $h_{t-j}^2$ measuring the forecast of the last period variance. The restriction of non-negative values for the parameters ($\omega, \alpha_i$ and $\beta_j$) it is important to ensure positive values for the conditional variance, which is $h_t^2 \geq 0$; otherwise, the model is not stable in variance. Moreover, the size of the two parameters $\alpha_i$ and $\beta_j$ determines the short-run dynamic volatility of the data, while the sum of their estimated values determines the persistence of volatility to a particular shock if $\alpha_i$ has a large and positive value; this indicates that the time series contains robust volatility clustering spikes that are short-lived. If $\beta_j$ has a large and positive value, indicating that the shocks' impact on the conditional variance lasts for a long time before dying out, so volatility is persistent.

The basic but most relevant GARCH process is the GARCH (1,1) model, also known as the generic or 'plain vanilla' GARCH model. Karmakar (2005) suggested the use of GARCH (1,1) to record conditional volatility in stock returns. The GARCH (1,1) model is written as follows, where p = 1 and q = 1; therefore, Equation (4) can be transformed into Equation (5) below:

$$h_t^2 = \omega + \alpha_1 \varepsilon_{t-1}^2 + \beta_1 h_{t-1}^2 \tag{5}$$

In Equation (5) ($\alpha_1$ and $\beta_1$) are the coefficients of the ARCH and GARCH terms, individually. Hence, ($\alpha$) (ARCH effect) estimates the response to shock and ($\beta$) (GARCH effect) measures the time it takes for any change to die away. As greater ($\alpha$) values illustrate higher sensitivity to new information, greater ($\beta$) values illustrate a greater amount of time for the change to die out. ($\alpha + \beta$) provide a measure of persistence of the relevant time series and thus higher values for ($\alpha + \beta$) should tend towards one and indicate greater persistence in volatility (Rastogi 2014). Hence, it must be mentioned that there are two cases that need to be considered, one when ($\alpha + \beta$) > 1 and the other where ($\alpha + \beta$) = 1 that will lead to an unstable GARCH model and an integrated process respectively. The first case implies that the GARCH model is non-stationary; the volatility will eventually explode as time goes to infinity. The second case is a restricted version of the standard GARCH model, which is well-known in the literature IGARCH model (Alexander 2001; Mittnik et al. 2007).

### 3.1.2. FIGARCH(1,d,1)

The FIGARCH model is represented in Equation (6) below.

$$h_t^2 = \omega + \left\{ 1 - |1 - \beta_1 L|^{-1}(1 - \phi_1 L)\Big|1 - L\Big|^d \right\}\epsilon_t^2 \tag{6}$$

Similar to the GARCH (1,1), Baillie et al. (1996) argued on the importance of ensuring a positive conditional variance of the FIGARCH(1,d,1) model. As such, all the parameters $\omega, \alpha, \beta$ must be positive. Moreover, $\alpha, \beta$ must be less than 1 and the sum of the coefficients $\alpha$ and $\beta$ must be $\leq 1$ otherwise, the model collapses, and it is not considered to be stable. In addition, the d parameter that captures the long memory process must be in the range of (0 to 0.5), if $0 < d < 0.5$ the series is stationary, if the $0.5 < d < 1$ the process is mean reverting as there is no long-run impact of innovation to future values. Hence, If the $d = 0$, the FIGARCH model collapses to the vanilla GARCH model and when the $d = 1$, it moves to an IGARCH model (Härdle and Mungo 2007; Salatas 2017). This research study is supported by the implementation of the well-known GARCH model in parallel to the FIGARCH model to examine if the dual shock affected Kuwait and the studied global markets in terms of their volatility performance. The analysis aims to identify if the markets exhibited a long memory process or if their behaviour was more in alignment with volatility clustering and persistence dynamics.

## 4. Research Findings

The analysis starts with a series of descriptive statistics to review the essential characteristics of the data. It continues with the analysis and discussion of the outcomes of the volatility models. The research study was supported by traditional time series tests that included a VAR (p) to identify the appropriate number of lags and implement the ADF, PP and KPSS tests for stationarity and robustness. Tables 1–4 below depicts the core outcomes from the descriptive statistics. The research findings indicate that the markets exhibited positive mean prices and returns over the period of study. Overall, the series are quite volatile as per the registered standard deviations, and the data are non-normal as common characteristics exhibited by financial time series. All markets (G7, E7, GCC, oil benchmarks) have positive values, indicating the existence of profit in returns. On the other hand, all markets' exhibited negative skewness and a leptokurtic shape, which means that the kurtosis value is more than 3 indicating a more peaked than a normal distribution with a long tail. The Jarque-Bera test for normality was significant at 1% significance level for all market returns confirming that the series were not normally distributed. The reason for the positive mean and the negative skewness with kurtosis can be justified by the market recovery process leading to higher prices than the decrease in prices experienced during the shock.

**Table 1.** G7 Descriptive Statistics.

| | CAC40 | DAX | DOW | FTSEE100 | FTSEE_MIB | NIK225 | SP_500 | SP_TSX |
|---|---|---|---|---|---|---|---|---|
| | **G7 Prices** | | | | | | | |
| Mean | 5289.547 | 12,392.04 | 25,228.47 | 6957.002 | 21,062.09 | 22,043.23 | 2930.174 | 16,200.59 |
| Std. Dev. | 667.7198 | 1624.917 | 5095.301 | 560.7568 | 2708.554 | 3758.534 | 698.8947 | 1911.047 |
| Skewness | 0.508426 | 0.323883 | 0.296818 | −0.854301 | 0.099389 | 0.497230 | 0.856064 | 0.809701 |
| Kurtosis | 3.006647 | 2.853291 | 2.488434 | 2.826973 | 2.453807 | 2.648013 | 2.965448 | 3.765438 |
| Jarque-Bera | 66.82441 | 28.50770 | 39.68649 | 190.5959 | 21.83284 | 71.91763 | 189.5179 | 207.3403 |
| Probability | 0.000000 | 0.000001 | 0.000000 | 0.000000 | 0.000018 | 0.000000 | 0.000000 | 0.000000 |
| | **G7 Returns** | | | | | | | |
| | CAC40R | DAXR | DOWR | FTSEE100R | FTSEE_MIBR | NIK225R | SP_500R | SP_TSXR |
| Mean | 0.000266 | 0.000242 | 0.000464 | 0.000103 | 0.000145 | 0.000266 | 0.000533 | 0.000307 |
| Std. Dev. | 0.011939 | 0.012177 | 0.011902 | 0.010442 | 0.014481 | 0.012344 | 0.011429 | 0.010126 |
| Skewness | −1.324078 | −0.967532 | −1.199691 | −1.078655 | −2.195639 | −0.209841 | −1.143895 | −2.099587 |
| Kurtosis | 19.75326 | 18.61260 | 30.27275 | 19.57829 | 29.25117 | 9.255081 | 26.31705 | 54.93787 |
| Jarque-Bera | 18,579.62 | 15,984.22 | 48,409.09 | 18,050.64 | 45,751.31 | 2538.265 | 35,451.01 | 175,355.1 |
| Probability | 0.000000 | 0.000000 | 0.000000 | 0.000000 | 0.000000 | 0.000000 | 0.000000 | 0.000000 |

Note: This table reports the summary statistics of daily prices and returns for the G7 stock markets. The sample period under consideration spans between 31 December 2015, and 9 December 2021. The Std. Dev., represents the prices and returns standard deviation. The Jarque-Bera for normality is included (the *p*-value at 1% significance level was considered, and the values are presented in the probability section).

**Table 2.** E7 Descriptive Statistics.

| | BRAZIL | BSE100 | INDONISIA | MEXICO | RUSSIA | SHANGHAI | BIST 100 |
|---|---|---|---|---|---|---|---|
| | **E7 Prices** | | | | | | |
| Mean | 86,362.76 | 11,327.99 | 5759.162 | 45,476.27 | 1223.424 | 3123.852 | 1060.451 |
| Std. Dev. | 22,883.80 | 2545.121 | 579.1687 | 4266.577 | 239.0116 | 278.9278 | 232.6517 |
| Skewness | −0.065500 | 0.978917 | −0.615834 | −0.664609 | 0.494991 | −0.042083 | 0.887424 |
| Kurtosis | 2.003040 | 3.666062 | 2.412220 | 2.850342 | 3.179866 | 2.215041 | 3.591154 |
| Jarque-Bera | 65.34168 | 276.3850 | 120.3633 | 115.6281 | 65.42743 | 40.27714 | 226.1583 |
| Probability | 0.000000 | 0.000000 | 0.000000 | 0.000000 | 0.000000 | 0.000000 | 0.000000 |
| | **E7 Returns** | | | | | | |
| | BRAZILR | BSE100R | INDONISIAR | MEXICOR | RUSSIAR | SHANGHAIR | TURKEYR |
| Mean | 0.000579 | 0.000510 | 0.000238 | 0.000113 | 0.000496 | 0.000024 | 0.000672 |
| Std. Dev. | 0.016670 | 0.010932 | 0.009776 | 0.010138 | 0.016161 | 0.011066 | 0.013238 |
| Skewness | −1.315873 | −1.719867 | −0.019816 | −0.597218 | −1.150530 | −1.103178 | −0.981545 |
| Kurtosis | 20.22241 | 28.47389 | 13.87721 | 7.797338 | 14.39119 | 11.26366 | 8.917113 |
| Jarque-Bera | 19,603.46 | 42,673.50 | 7641.196 | 1578.490 | 8722.245 | 4724.665 | 2510.092 |
| Probability | 0.000000 | 0.000000 | 0.000000 | 0.000000 | 0.000000 | 0.000000 | 0.000000 |

Note: This table reports the summary statistics of daily prices and returns for E7 stock markets. The research sample under consideration spans between 31 December 2015, and 9 December 2021. The Std. Dev., represents the prices and returns standard deviation. The Jarque-Bera for normality is included (the *p*-value at 1% significance level was considered with values presented in the probability section).

**Table 3.** GCC Descriptive Statistics.

| | Bahrain | Dubai | Kuwait | Qatar | Saudi | Oman |
|---|---|---|---|---|---|---|
| **GCC prices** | | | | | | |
| Mean | 1373.505 | 2926.980 | 5264.773 | 2936.463 | 7958.605 | 4526.922 |
| Std. Dev. | 156.7539 | 471.4857 | 734.5464 | 320.7747 | 1405.781 | 778.0595 |
| Skewness | 0.429677 | −0.260928 | 0.492258 | 0.166400 | 0.946562 | 0.383964 |
| Kurtosis | 2.733524 | 2.393174 | 2.813190 | 3.149955 | 3.599616 | 1.752582 |
| Jarque-Bera | 52.31390 | 41.39697 | 64.89441 | 8.610821 | 254.8462 | 138.6701 |
| Probability | 0.000000 | 0.000000 | 0.000000 | 0.013495 | 0.000000 | 0.000000 |
| **GCC returns** | | | | | | |
| | Bahrain | Dubai | Kuwait | Qatar | Saudi | Oman |
| Mean | 0.000248 | 0.0000152 | 0.000293 | 0.000182 | 0.000296 | −0.000193 |
| Std. Dev. | 0.005143 | 0.010979 | 0.007999 | 0.009559 | 0.010416 | 0.005147 |
| Skewness | −1.506418 | −0.653869 | −3.190621 | −1.263180 | −1.187625 | −0.940610 |
| Kurtosis | 21.43615 | 14.44610 | 40.38432 | 18.81032 | 14.48466 | 16.67683 |
| Jarque-Bera | 22,537.57 | 8571.714 | 92,890.71 | 16,555.86 | 8882.749 | 12,309.25 |
| Probability | 0.000000 | 0.000000 | 0.000000 | 0.000000 | 0.000000 | 0.000000 |

Note: This table reports the summary statistics of daily prices and returns for GCC stock markets, The research sample under consideration spans between 31 December 2015, and 9 December 2021. The Std. Dev., represents the prices and returns standard deviation. The Jarque-Bera for normality is included (the *p*-value at 1% significance level was considered with values presented in the probability section).

**Table 4.** Oil Benchmarks Descriptive Statistics.

| | BRENT | DUBAI | OPEC | WTI |
|---|---|---|---|---|
| **Oil Prices** | | | | |
| Mean | 58.15781 | 56.25103 | 56.12941 | 53.72606 |
| Std. Dev. | 13.41835 | 13.44321 | 14.45190 | 12.57164 |
| Skewness | −0.250214 | −0.278254 | −0.420205 | −0.243291 |
| Kurtosis | 2.535989 | 2.446104 | 2.696566 | 3.204172 |
| Jarque-Bera | 30.09810 | 39.84141 | 51.59412 | 17.99466 |
| Probability | 0.000000 | 0.000000 | 0.000000 | 0.000124 |
| **Oil Returns** | | | | |
| | BRENTR | DUBAIR | OPECR | WTIR |
| Mean | 0.000474 | 0.000513 | 0.000568 | 0.000419 |
| Std. Dev. | 0.026910 | 0.026348 | 0.026952 | 0.032927 |
| Skewness | −2.748229 | −0.969671 | −1.822894 | −1.058766 |
| Kurtosis | 58.55986 | 21.97041 | 40.09242 | 37.67027 |
| Jarque-Bera | 201,313.3 | 23,484.92 | 89,715.24 | 77,920.53 |
| Probability | 0.000000 | 0.000000 | 0.000000 | 0.000000 |

Note: This table reports the summary statistics of daily prices and returns for the studied Oil benchmarks are represented by the Brent, Dubai, OPEC and WTI indices. The Std. Dev., represents the prices and returns standard deviation. The Jarque-Bera for normality is included (the *p*-value at 1% significance level was considered with values presented in the probability section).

*4.1. Stationarity Findings*

The series stationarity properties were examined by three well-known tests, the ADF (Augmented Dickey-Fuller), the PP (Phillips-Perron) and the KPSS (Kwiatkowski- Phillips-



Schmidt- Shin), due to significant levels of criticism associated with the performance of the ADF the PP and KPSS were implemented for robustness (Asteriou and Hall 2011; Taheri 2014). Moreover, the random walk with drift approach is used because we do not assume the existence of a pure random walk in line with the work done by Alshogeathri (2011). Tables 5–8 present the outcomes of the unit root test. The tests show that the series (G7, E7, GCC indexes and oil benchmarks) are non-stationary in levels but are stationary at 1% level in returns. These results align with common research findings associated with the study of financial time series.

**Table 5.** G7 Unit Root Testing.

| | Returns | | | | PricesPrices | | | |
| --- | --- | --- | --- | --- | --- | --- | --- | --- |
| **G7** | **ADF** | **PP** | **KPSS *** | **lags** | **ADF** | **PP** | **KPSS *** | **Lags** |
| CAC−40 | −39.1311 (0.0000) | −39.1992 (0.0000) | 0.06288 (0.739000) | 0 | 0.905493 (0.7869) | 1.102971 (0.7167) | 2.44178 (0.739000) | 1 |
| DAX | −39.8231 (0.0000) | −39.8574 (0.0000) | 0.043763 (0.739000) | 0 | −1.25996 (0.6500) | −1.35873 (0.6038) | 2.635266 | 1 |
| DOW | −12.0682 (0.0000) | −47.4828 (0.0001) | 0.029875 (0.739000) | 9 | −0.87707 (0.7958) | −0.57271 (0.874) | 4.208865 | 10 |
| FTSEE−100 | −39.8899 (0.0000) | −39.8936 (0.0000) | 0.062976 (0.739000) | 0 | −2.47008 (0.1231) | −2.49244 (0.1175) | 0.601998 | 1 |
| FTSEE_MIB | −26.3578 (0.0000) | −42.0435 (0.0000) | 0.083753 (0.739000) | 2 | 1.473205 (0.5472) | 1.737985 (0.4118) | 1.850557 | 1 |
| Nikkei−225 | −40.3251 (0.0000) | −40.3212 (0.0000) | 0.057721 (0.739000) | 0 | 0.895518 (0.7901) | 0.914543 (0.784) | 3.575692 | 1 |
| S&P500 | −11.9827 (0.0000) | −48.3939 (0.0001) | 0.070294 (0.739000) | 9 | 0.213192 (0.9734) | 0.541773 0.9881 | 4.211441 | 10 |
| S&P-TSX | −12.8414 (0.0000) | −46.4586 (0.0001) | 0.04685 (0.739000) | 7 | −1.52392 (0.5214) | −1.12548 (0.7077) | 3.024149 | 8 |

* Note: the *p*-values are shown in parentheses; there is no *p*-value for KPSS; therefore the 1% significance level was considered for the test at a value of 0.739000. The main indices for the G7 stock markets are presented, and the results for the three stationarity tests with *p*-values in brackets and the number of required lags to estimate the tests are reported in the table.

**Table 6.** E7 Unit Root Tests.

| | Returns | | | | Prices | | | |
| --- | --- | --- | --- | --- | --- | --- | --- | --- |
| **E7** | **ADF** | **PP** | **KPSS *** | **Lags** | **ADF** | **PP** | **KPSS *** | **Lags** |
| TURKEY | −38.6391 (0.0000) | −38.765 (0.0000) | 0.163642 | 0 | 1.956619 (0.9999) | 1.302195 (0.9987) | 3.371059 | 1 |
| BRAZIL | −45.9358 (0.0001) | −45.3963 (0.0001) | 0.125707 | 1 | −1.7012 (0.4305) | −1.79155 (0.385) | 4.313622 | 2 |
| Indonesia | −37.2066 (0.0000) | −37.2674 (0.0000) | 0.088002 | 0 | −2.04716 (0.2667) | −2.1554 (0.2231) | 1.020901 | 1 |
| Mexico | −37.2846 (0.0000) | −37.2316 (0.0000) | 0.100463 | 0 | −1.83143 (0.3654) | −1.80461 (0.3786) | 0.785645 | 1 |
| Russia | −40.1799 (0.0000) | −40.1864 (0.0000) | 0.047403 | 0 | −1.68253 (0.4400) | −1.76879 (0.3964) | 3.439044 | 1 |
| BSE100 | −16.6838 (0.0000) | −40.3195 (0.0000) | 0.104646 | 6 | 0.46492 (0.9855) | 0.408541 (0.9833) | 3.430955 | 1 |
| Shanghai | −41.4445 (0.0000) | −41.3982 (0.0000) | 0.156396 | 0 | −2.21347 (0.2016) | −2.22213 (0.1985) | 1.131125 | 1 |

* Note: the *p*-values are shown in parentheses; there is no *p*-value for KPSS; therefore, the 1% significance level was considered for the test at a value of 0.739000. The main indices for the E7 stock markets are presented, and the results for the three stationarity tests with *p*-values in brackets and the number of required lags to estimate the tests are reported in the table.

**Table 7.** GCC Unit Root test.

| GCC | Returns | | | | Prices | | | |
| | ADF | PP | KPSS * | Lags | ADF | PP | KPSS * | Lags |
|---|---|---|---|---|---|---|---|---|
| Bahrain | 23.9287 (0.0000) | 36.3923 (0.0000) | 0.163642 | 3 | −0.40241 (0.9064) | −0.1814 (0.9383) | 3.38354 | 7 |
| Dubai | 35.5196 (0.0000) | 36.5589 (0.0000) | 0.127087 | 1 | −1.2989 (0.6321) | −1.52308 (0.5218) | 3.162082 | 2 |
| Kuwait | 33.5391 (0.0000) | 33.8347 (0.0000) | 0.073448 | 5 | −0.40294 (0.9063) | −0.55905 (0.8769) | 3.433574 | 6 |
| Qatar | 37.5455 (0.0000) | 37.7328 (0.0000) | 0.086213 | 1 | −0.8174 0.8135 | −1.10685 (0.7152) | 2.586876 | 2 |
| Saudi Arabia | 34.7529 (0.0000) | 35.0339 (0.0000) | 0.085712 | 5 | −0.43677 (0.9004) | −0.50296 (0.8882) | 3.124048 | 2 |
| Oman | −30.58239 (0.0000) | −30.60818 (0.0000) | 0.149798 | 1 | −1.027663 (0.7453) | −1.056701 (0.7453) | 4.434236 | 2 |

* Note: the *p*-values are shown in parentheses; there is no *p*-value for KPSS; therefore, the 1% significance level was considered for the test at a value of 0.739000. The main indices for the GCC stock markets are presented, and the results for the three stationarity tests with *p*-values in brackets and the number of required lags to estimate the tests are reported in the table.

**Table 8.** Oil benchmarks Unit Root Tests.

| OIL | Returns | | | | Price | | | |
| | ADF | PP | KPSS * | Lags | ADF | PP | KPSS * | Lags |
|---|---|---|---|---|---|---|---|---|
| Brent | 38.1946 (0.0000) | −38.217 (0.0000) | 0.07235 (0.739000) | 0 | −1.98352 (0.2943) | −2.06769 (0.2581) | 0.75734 | 1 |
| Dubai | 39.7635 (0.0000) | 39.8059 (0.0000) | 0.082546 (0.739000) | 0 | −2.07198 (0.2563) | −2.06098 (0.2609) | 0.911146 | 1 |
| OPEC | 37.6637 (0.0000) | 38.5049 (0.0000) | 0.069871 (0.739000) | 2 | −1.84347 (0.3596) | −1.97708 (0.2972) | 0.892014 | 1 |
| WTI | −30.673 (0.0000) | 39.1462 (0.0000) | 0.04562 (0.739000) | 0 | −1.96184 (0.304) | −2.11726 (0.2379) | 0.668896 | 4 |

* Note: the *p*-values are shown in parentheses; there is no *p*-value for KPSS; therefore, the 1% significance level was considered for the test at a value of 0.739000. The main indices for the Oil Benchmarks indices are presented, and the results for the three stationarity tests with *p*-values in brackets and the number of required lags to estimate the tests are reported in the table.

*4.2. Volatility Findings*

4.2.1. G7 Findings

The outcomes of the GARCH (1,1) for the G7 countries reveal that the model was quite efficient in capturing volatility dynamics as all associated *p*-values were significant at 1% level. The alpha coefficient representing recent news related to current market volatility spikes is in the range of $\alpha$ = (0.113911, 0.237453), and the beta representing persistence is in the range of $\beta$ = (0.728516, 0.848021). The DAX 30 exhibited the lowest volatility spikes with the highest persistence, and the S&P 500 had the highest volatility spikes with the lowest persistence. The range of the alpha and beta ($\alpha+\beta$) for all indexes was in the range of 0.97039 and 0.95003, with the highest values associated with the Canadian index S&P-TSX, and the lowest with the Japanese index NIKKEI-225. All the markets registered high persistence levels, with all the markets $\alpha+\beta$ being above 0.95. Moreover, the $\alpha+\beta$ coefficients are less than one for all the series under study illustrating that the GARCH model is stationary and stable. The GARCH $\beta$ coefficient is larger than the coefficient of $\alpha$ for all indexes, which signifies the existence of significant clustering behaviour across markets, as shown in Figure 5. Moreover, the volatility effects for the G7 markets lasted

between 24 to 14 days, with the longest effects registered by the Canadian index S&P-TSX, and the shortest by the Japanese index NIKKEI-225, as illustrated in Table 9 below.

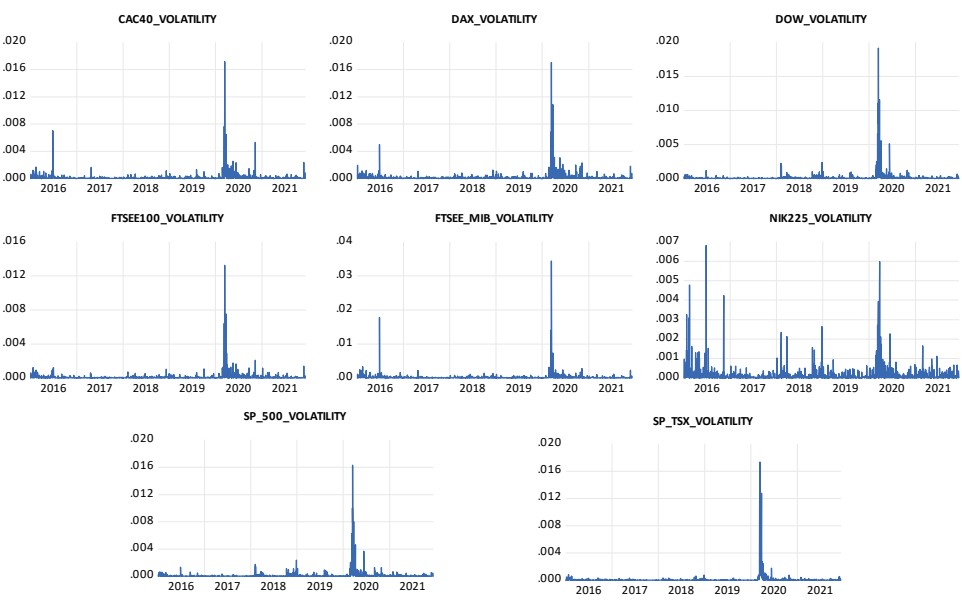

**Figure 5.** G7 volatility clustering (sources Data Stream).

**Table 9.** G7 Volatility test.

| | | G7 COUNTRIES | | | | | | | |
|---|---|---|---|---|---|---|---|---|---|
| | | **S&P-TSX** | **Dow-** | **FTSE-MIB** | **DAX** | **CAC-40** | **FTSEE-100** | **S&P500** | **NIKKEI-225** |
| **GARCH(1,1)** | w | $2.18 \times 10^{-6}$ (0.0000) | $4.17 \times 10^{-6}$ (0.0000) | $7.29 \times 10^{-6}$ (0.0000) | $4.96 \times 10^{-6}$ (0.0000) | $6.46 \times 10^{-6}$ (0.0000) | $4.05 \times 10^{-6}$ (0.0000) | $4.34 \times 10^{-6}$ (0.0000) | $7.22 \times 10^{-6}$ (0.0000) |
| | α | 0.20666 (0.0000) | 0.221959 (0.0000) | 0.140486 (0.0000) | 0.113911 (0.0000) | 0.179019 (0.0000) | 0.122174 (0.0000) | 0.236205 (0.0000) | 0.11597 (0.0000) |
| | β | 0.763728 (0.0000) | 0.743244 (0.0000) | 0.824016 (0.0000) | 0.848021 (0.0000) | 0.777263 (0.0000) | 0.831288 (0.0000) | 0.73032 (0.0000) | 0.834064 (0.0000) |
| | α+β | 0.97039 | 0.96520 | 0.964502 | 0.96193 | 0.95628 | 0.95346 | 0.966525 | 0.95003 |
| | Half-life (days) | 24 | 20 | 20 | 18 | 16 | 15 | 21 | 14 |
| **FIGARCH(1,1)** | w | $2.32 \times 10^{-6}$ (0.0005) | $4.94 \times 10^{-6}$ (0.0000) | $7.73 \times 10^{-6}$ (0.0000) | $5.72 \times 10^{-6}$ (0.0000) | $8.33 \times 10^{-6}$ (0.0000) | $3.37 \times 10^{-6}$ 0.0061 | $4.95 \times 10^{-6}$ 0.0000 | $1.57 \times 10^{-5}$ 0.0001 |
| | α | 0.169839 (0.0454) | 0.015178 (0.8326) | 0.183796 (0.0001) | 0.067884 (0.2082) | −0.056357 (0.4219) | 0.295065 (0.0027) | 0.043248 (0.5787) | −0.194230 (0.2318) |
| | β | 0.513544 (0.0000) | 0.461361 (0.0000) | 0.549464 (0.0000) | 0.476279 (0.0000) | 0.360457 (0.0001) | 0.489335 (0.0002) | 0.435936 (0.0000) | −0.045886 (0.7927) |
| | d | 0.642651 (0.0000) | 0.66737 (0.0000) | 0.542572 (0.0000) | 0.525644 (0.0000) | 0.564086 (0.0000) | 0.380181 (0.0000) | 0.633111 (0.0000) | 0.298796 (0.0000) |

Note: table illustrates the GARCH (1,1) and FIGARCH (1,1). The GARCH (1,1) parameters (α) estimates the response to shock, (β) measures the time it takes for any change to die away, (α + β) provide a measure of persistence of the relevant time series. FIGARCH (1,1) parameter (d) captures the long memory process. The *p*-values are shown in parentheses. The greater (α) values illustrate higher sensitivity to new information, and greater (β) values illustrate a greater amount of time for the change to die out. (α + β) provide a measure of the persistence of the relevant time series, and thus higher values for (α + β) should tend towards one and indicate greater persistence in volatility. Moreover, α, β must be less than 1 and the sum of the coefficients α and β must be ≤ 1; otherwise, the model collapses, and it is not considered to be stable. The d parameter for the FIGARCH model captures the long memory process must be in the range of (0 to 0.5), if 0 < d < 0.5 the series is stationary, if the 0.5 < d < 1 the process is mean reverting as there is no long-run impact of innovation to future values.

The outcomes of the FIGARCH (1,1) model show that not all indexes are significant. For example, the CAC40 and NIKKIE−225 coefficients were not stable, as according to Baillie et al. (1996), the model requires all coefficients to be positive. On the other hand, S&P-TSX, DOW, FTSE-MIB, DAX, S&P500, recorded a *d* coefficient with the following values *d* = (0.642651), (0.66737), (0.551935), (0.525644), (0.633111) that according to Härdle and Mungo (2007), when the d coefficient is between 0.5 < d < 1 the process is mean

reverting as there is no long-run impact of innovation to future values. The only index that complied with the model limitations was the FTSE 100 d = (0.380181), indicating a long memory process.

### 4.2.2. E7 Volatility Findings

The outcomes of the GARCH (1,1) for E7 countries present significant results. All the *p*-values are significant at 1% level. The alpha coefficient representing recent news related to current market volatility spikes, is in the range of α = (0.064525, 0.118014). The highest was the Mexican index BOLSA, and the lowest was the Chinese index Shanghai all composite. The beta coefficient capturing the persistence effects is in the range of β = (0.838354, 0.923037). The highest value was recorded by the Chinese index Shanghai, and the lowest value by the Turkish index BIST100. Moreover, the alpha and beta (α+β) result was in the range of 0.911652 to 0.987562. The highest value was associated with the Chinese index and the lowest with the Turkish index BIST100. All market values are above 0.90, indicating a high volatility level of persistence of returns. Moreover, all indexes coefficients of α+β are less than 1, which illustrates that all coefficients are stationary and stable. The GARCH effect coefficient of β is larger than the value associated with the α coefficient, which signifies that clustering behaviour is present, as shown in Figure 6. Volatility lasting effects are reflected in the half-life volatility for E7 markets that ranged between 8 to 56 days, with the most prolonged effects recorded in the Chinese index and the shortest associated with the Turkish index BIST100, as represented in Table 10 below.

**Table 10.** E7 Volatility test.

| | | BOVESPA | BIST 100 | RTS INDEX | BOLSA | IDEX | SHANGHAI | BSE 100 |
|---|---|---|---|---|---|---|---|---|
| | | **E7 COUNTRIES** | | | | | | |
| GARCH(1,1) | w | $1.15 \times 10^{-5}$ (0.0000) | $1.55 \times 10^{-5}$ 0.0002 | $4.61 \times 10^{-6}$ (0.0000) | $3.72 \times 10^{-6}$ (0.0000) | $3.74 \times 10^{-6}$ (0.0000) | $1.50 \times 10^{-6}$ (0.0000) | $2.08 \times 10^{-6}$ (0.0000) |
| | α | 0.090865 (0.0000) | 0.073298 (0.0000) | 0.073949 (0.0000) | 0.118014 (0.0000) | 0.104727 (0.0000) | 0.064525 (0.0000) | 0.091972 (0.0000) |
| | β | 0.855811 (0.0000) | 0.838354 (0.0000) | 0.908133 (0.0000) | 0.843706 (0.0000) | 0.850918 (0.0000) | 0.923037 (0.0000) | 0.887281 (0.0000) |
| | α+β | 0.946676 | 0.911652 | 0.982082 | 0.96172 | 0.955645 | 0.987562 | 0.979253 |
| | Half-life (days) | 13 | 8 | 39 | 18 | 16 | 56 | 34 |
| FIGARCH(1,1) | w | $6.97 \times 10^{-5}$ (0.0000) | $4.31 \times 10^{-5}$ 0.0022 | $3.13 \times 10^{-6}$ 0.0070 | $6.44 \times 10^{-6}$ 0.0019 | $4.47 \times 10^{-6}$ 0.0039 | $9.79 \times 10^{-7}$ 0.4686 | $3.56 \times 10^{-6}$ (0.0012) |
| | α | −0.582253 (0.0000) | 0.008691 (0.9622) | −0.094854 (0.1173) | −0.066332 (0.6128) | 0.351761 (0.0017) | −0.023439 (0.0000) | 0.120816 (0.0614) |
| | β | −0.505079 (0.0005) | 0.148466 (0.4605) | 0.90082 (0.0000) | 0.17695 (0.2282) | 0.506512 (0.0000) | 0.917799 (0.0000) | 0.498124 (0.0000) |
| | d | 0.186585 (0.0000) | 0.196344 (0.0000) | 0.985848 (0.0000) | 0.359226 (0.0000) | 0.310872 (0.0000) | 0.996757 (0.0000) | 0.48343 (0.0000) |

Note: table illustrates the GARCH (1,1) and FIGARCH (1,1). The GARCH (1,1) parameters (α) estimates the response to shock, (β) measures the time it takes for any change to die away, (α + β) provide a measure of persistence of the relevant time series. FIGARCH (1,1) parameter (d) captures the long memory process. The *p*-values are shown in parentheses. The greater (α) values illustrate higher sensitivity to new information, and greater (β) values illustrate a greater amount of time for the change to die out. (α + β) provide a measure of the persistence of the relevant time series, and thus higher values for (α + β) should tend towards one and indicate greater persistence in volatility. Moreover, α, β must be less than 1 and the sum of the coefficients α and β must be ≤ 1; otherwise, the model collapses, and it is not considered to be stable. The d parameter for the FIGARCH model captures the long memory process must be in the range of (0 to 0.5), if 0 < *d* < 0.5 the series is stationary, if the 0.5 < *d* < 1 the process is mean reverting as there is no long-run impact of innovation to future values.

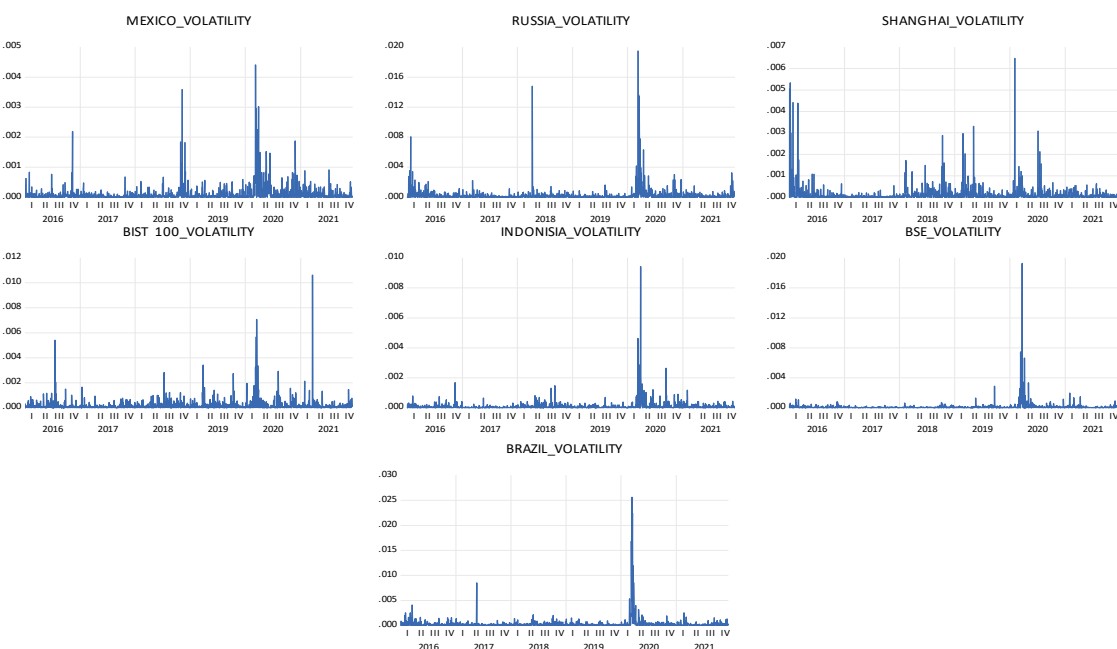

**Figure 6.** E7 volatility clustering (sources Data Stream).

The outcome of the FIGARCH (1,1) shows a lack of a long memory process. As RTS INDEX, BOLSA, and SHANGHAI show negative volatility spikes $\alpha$ = ($-0.094854$), ($-0.066332$), ($-0.023439$), and BOVESPA shows negative volatility spikes and persistence $\alpha$ = ($-0.582253$) and $\beta$ = ($-0.505079$). On the other hand, BIST 100, IDEX, BSE 100 D = 0.196344, 0.310872, 0.48343 has long memory volatility as the d is positive and <0.5 (Härdle and Mungo 2007).

### 4.2.3. GCC Volatility Findings

The outcomes of the GARCH (1,1) for the GCC countries present significant results. All the *p*-values are significant at the 1% level. The alpha coefficient reveals that Kuwait recorded the highest volatility spikes, with Dubai associated with the lowest levels. The beta coefficient representing persistence is in the $\beta$ = (0.647571, 0.853152), with Bahrain registering the lowest persistence and Dubai exhibiting the highest persistence. Hence, the alpha and beta ($\alpha$+$\beta$) KSE, DUBAI, QATAR, TASI and Oman range was (0.904941, 0.972619) above 0.90, indicating high volatility levels of persistence of stock returns. Conversely, Bahrain recorded (0.796208) low volatility compared to the rest of the GCC. The alpha and beta ($\alpha$+$\beta$) results for all markets are less than one, illustrating that coefficients are stationary and stable. The GARCH effect coefficient of $\beta$ is larger than the coefficient of $\alpha$ for all indexes, which signifies clustering behaviour as shown in Figure 7. In addition, lasting volatility effects for the GCC markets ranged between 4 and 25 days, with the highest persistence registered by the Qatar index and the lowest with the Bahrain index as represented in Table 11 below.

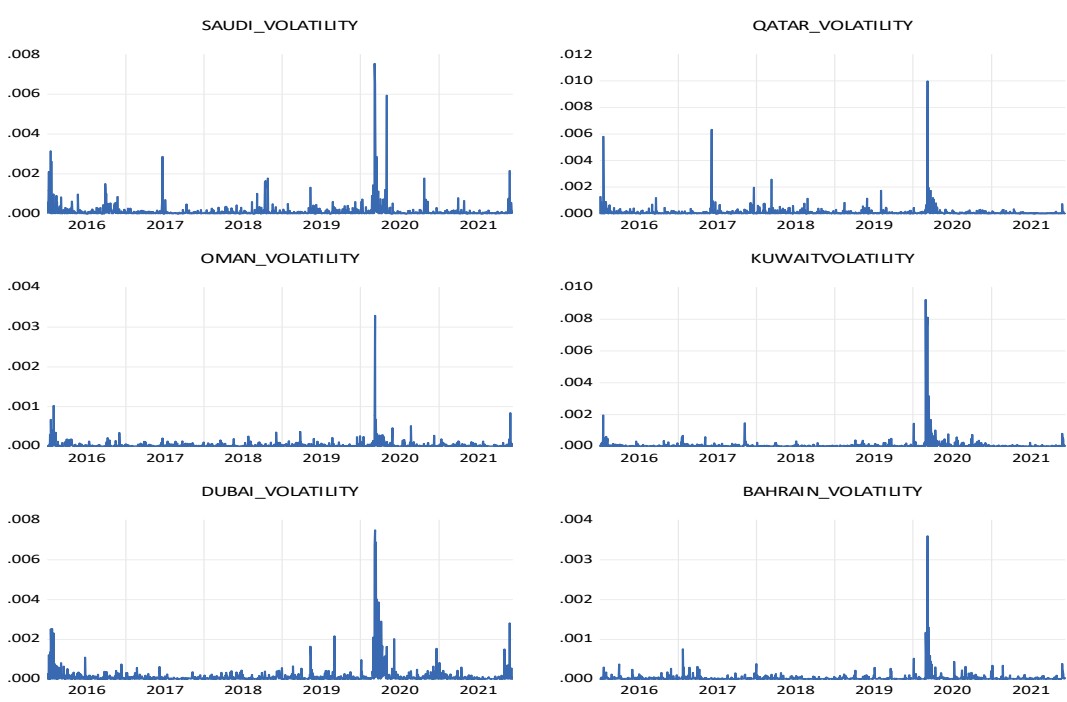

**Figure 7.** GCC volatility clustering (sources Data Stream).

**Table 11.** GCC Volatility test.

| | | GCC COUNTRIES | | | | | |
|---|---|---|---|---|---|---|---|
| | | **Kuwait** | **Dubai** | **Qatar** | **Saudi** | **Bahrain** | **Oman** |
| **GARCH (1,1)** | w | $2.47 \times 10^{-6}$ (0.0000) | $4.16 \times 10^{-6}$ (0.0000) | $3.15 \times 10^{-6}$ (0.0000) | $3.60 \times 10^{-6}$ (0.0000) | $4.68 \times 10^{-6}$ (0.0000) | $2.40 \times 10^{-6}$ (0.0000) |
| | α | 0.156627 (0.0000) | 0.104032 (0.0000) | 0.131418 (0.0000) | 0.147074 (0.0000) | 0.148637 (0.0000) | 0.155814 (0.0000) |
| | β | 0.813056 (0.0000) | 0.853152 (0.0000) | 0.841201 (0.0000) | 0.821419 (0.0000) | 0.647571 (0.0000) | 0.749127 (0.0000) |
| | α+β | **0.969683** | **0.957184** | **0.972619** | **0.968493** | **0.796208** | **0.904941** |
| | Half-life (days) | **23** | **16** | **25** | **22** | **4** | **8** |
| **FIGARCH (1,1)** | w | $3.14 \times 10^{-6}$ (0.0000) | $3.01 \times 10^{-6}$ (0.0000) | $3.21 \times 10^{-6}$ (0.0000) | $1.61 \times 10^{-6}$ (0.0000) | $4.15 \times 10^{-6}$ (0.0000) | $1.87 \times 10^{-6}$ (0.0000) |
| | α | 0.005245 (0.8741) | 0.297341 (0.0000) | 0.156179 (0.0000) | −0.09125 0.154 | 0.771257 (0.0000) | 0.155616 (0.0028) |
| | β | 0.658701 (0.0000) | 0.713121 (0.0000) | 0.609766 (0.0000) | 0.901126 (0.0000) | 0.647162 (0.0000) | 0.560132 (0.0000) |
| | d | 0.765222 (0.0000) | 0.60103 (0.0000) | 0.615344 (0.0000) | 1.175244 (0.0000) | 0.03101 (0.0360) | 0.603413 (0.0000) |

Note: table illustrates the GARCH (1,1) and FIGARCH (1,1). The GARCH (1,1) parameters (α) estimates the response to shock, (β) measures the time it takes for any change to die away, (α + β) provide a measure of persistence of the relevant time series. FIGARCH (1,1) parameter (d) captures the long memory process. The *p*-values are shown in parentheses. The greater (α) values illustrate higher sensitivity to new information, and greater (β) values illustrate a greater amount of time for the change to die out. (α + β) provide a measure of the persistence of the relevant time series, and thus higher values for (α + β) should tend towards one and indicate greater persistence in volatility. Moreover, α, β must be less than 1 and the sum of the coefficients α and β must be ≤ 1; otherwise, the model collapses, and it is not considered to be stable. The d parameter for the FIGARCH model captures the long memory process must be in the range of (0 to 0.5), if 0 < *d* < 0.5 the series is stationary if the 0.5 < *d* < 1 the process is mean reverting as there is no long-run impact of innovation to future values.

The outcome of the FIGARCH (1,1) shows no evidence of a long memory process, and the model was not stable for the regions as the Saudi TASI has negative volatility spikes α = (−0.09125) and d (1.175244) more than 1. Moreover, KSE, DUBAI, Qatar and Oman *d* = (0.765222), (0.60103), (0.615344) and (0.603413) indicating the lack of evidence supporting

the existence of a long memory process. Finally, Bahrain exhibited different behaviour compared to the other GCC countries with $d$ = (0.03101), as it revealed a positive $d < 0.5$.

### 4.2.4. Oil Benchmarks Volatility Findings

The outcomes of the GARCH (1,1) for oil benchmarks present significant results. All the *p*-values are significant at the 1% level. Hence, the alpha coefficient representing recent news related to current market volatility spikes is in the range of $\alpha$ = (0.116171, 0.170954), and the beta representing persistence is in the range of $\beta$ = (0.82298, 0.868401). The results of alpha and beta ($\alpha+\beta$) Brent, Dubai, OPEC, and WTI (0.984572, 0.975747, 0.993934 and 0.98804) are less than one which illustrates that coefficients are stable. The GARCH $\beta$ coefficient is larger than the $\alpha$ for all indexes, which signifies clustering behaviour as shown in Figure 8. The volatility lasting for oil benchmarks markets range was 29 to 114 days. The longest was OPEC, and the shortest was Dubai as represented in Table 12 below.

**Table 12.** Oil Benchmarks Volatility test.

| | | OIL PRICES | | | |
|---|---|---|---|---|---|
| | | **Brent** | **Dubai** | **OPEC** | **WTI** |
| **GARCH(1,1)** | **w** | $1.26 \times 10^{-5}$ (0.0000) | $1.82 \times 10^{-5}$ (0.0000) | $1.03 \times 10^{-5}$ (0.0000) | $2.00 \times 10^{-5}$ (0.0000) |
| | $\alpha$ | 0.116171 (0.0000) | 0.118827 (0.0000) | 0.170954 (0.0000) | 0.123824 (0.0000) |
| | $\beta$ | 0.868401 (0.0000) | 0.85692 (0.0000) | 0.82298 (0.0000) | 0.864216 (0.0000) |
| | $\alpha+\beta$ | 0.984572 | 0.975747 | 0.993934 | 0.98804 |
| | Half-life (days) | 45 | 29 | 114 | 58 |
| **FIGARCH(1,1)** | **W** | $9.09 \times 10^{-6}$ (0.0000) | $1.19 \times 10^{-5}$ (0.0000) | $8.20 \times 10^{-6}$ (0.0000) | $1.30 \times 10^{-5}$ (0.0000) |
| | $\alpha$ | −0.018239 (0.7201) | 0.001973 (0.9781) | −0.001668 (0.9616) | 0.00685 0.1143 |
| | $\beta$ | 0.881135 (0.0000) | 0.869669 (0.0000) | 0.85176 (0.0000) | 0.870597 (0.0000) |
| | d | 1.034108 (0.0000) | 1.012828 (0.0000) | 1.037388 (0.0000) | 1.010829 (0.0000) |

Note: table illustrates the GARCH (1,1) and FIGARCH (1,1). The GARCH (1,1) parameters ($\alpha$) estimates the response to shock, ($\beta$) measures the time it takes for any change to die away, ($\alpha + \beta$) provide a measure of persistence of the relevant time series. FIGARCH (1,1) parameter (d) captures the long memory process. The *p*-values are shown in parentheses. The greater ($\alpha$) values illustrate higher sensitivity to new information, and greater ($\beta$) values illustrate a greater amount of time for the change to die out. ($\alpha + \beta$) provide a measure of the persistence of the relevant time series, and thus higher values for ($\alpha + \beta$) should tend towards one and indicate greater persistence in volatility. Moreover, $\alpha$, $\beta$ must be less than 1 and the sum of the coefficients $\alpha$ and $\beta$ must be $\leq$ 1; otherwise, the model collapses, and it is not considered to be stable. The d parameter for the FIGARCH model captures the long memory process must be in the range of (0 to 0.5), if $0 < d < 0.5$ the series is stationary, if the $0.5 < d < 1$ the process is mean reverting as there is no long-run impact of innovation to future values.

The outcomes of the FIGARCH (1,1) for oil benchmarks illustrate that all the benchmarks' *d* parameter is >1. Moreover, Brent and OPEC have a negative ARCH ($\alpha$), demonstrating that the FIGARCH (1,1) collapsed.

*Diagnostic Tests*[2]

The GARCH (1,1) model showed that the residuals were homoscedastic. The heteroscedasticity test (ARCH-LM) where *the null hypothesis of no heteroscedasticity cannot be rejected* for G7, E7, GCC and the oil benchmarks. Our results are robust as all the studied markets did not exhibit serial correlation and no heteroscedasticity effects.

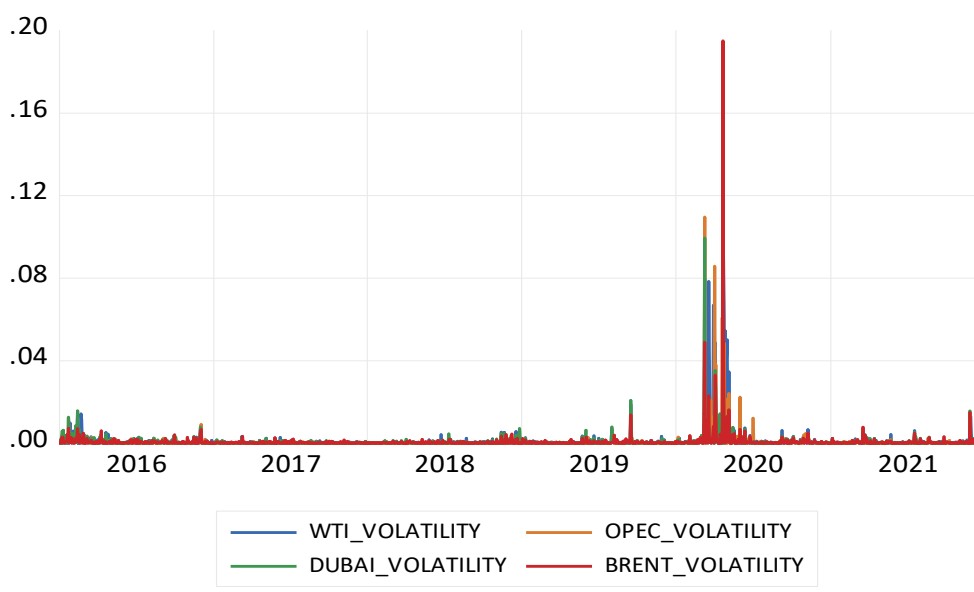

**Figure 8.** Oil Benchmarks volatility clustering (sources Data Stream).

## 5. Discussion

This study aimed to analyse the impact of a dual shock that took place in 2020 as the world faced the effects of the global health crisis and the oil prices war between the Kingdom of Saudi Arabia and Russia with a focused approach on implications for the Kuwait stock market. The study was enriched by including the world's most significant markets: the G7, E7, and the GCC and oil Benchmarks WTI, Brent, Dubai and OPEC to help us understand to what extent a small oil-dependent economy like Kuwait was affected by unfolding events. The analysis was supported by econometric modelling seeking to understand volatility persistency and long-memory processes by implementing the GARCH(1,1) and FIGARCH(1,1) models.

The core research findings suggest that the Kuwait index experienced a similar impact during the highlighted period to the Saudi index. The results show interesting insights, as they clearly illustrate that the economic models of Saudi Arabia and Kuwait share significant reliance on the oil market. Moreover, in line with early research studies examining Kuwait's stock exchange (Al-Shami and Ibrahim 2013; Al Hayky and Naim 2016; Merza and Almusawi 2016; Kisswani and Elian 2017; Alshihab and Al Shammari 2020; Alotaibi and Morales 2022) the outcomes of this research study support the evidence that the Kuwait stock market has a positive relation with oil volatility. This result is not surprising due to the country's heavy reliance on oil. Furthermore, and in line with Alotaibi and Morales (2022) research findings, the GARCH (1,1) model offered the best estimates as it captured the volatility persistence of the G7 markets. Consequently, the GARCH (1,1) efficiently captured the volatility persistence for G7, E7, GCC, and oil benchmarks (WTI, Brent, OPEC, Dubai). On the other hand, the FIGARCH (1,1) did not offer significant evidence of long memory processes affecting the analysed markets except for the cases of FTSE 100 in G7 countries, BIST 100, IDEX, BSE 100 in E7 countries, and Bahrain in GCC countries. In the case of oil benchmarks, the model collapsed.

In contrast, the research study developed by Bentes (2021) showed evidence on how the FIGARCH model worked well for all G7 markets by following the d component for the FIGARCH boundaries $0 < d < 1$. Our research differentiates from Bentes (2021) study as our boundaries align with those of Härdle and Mungo (2007) boundaries for the FIGARCH; the *d* component must be positive and equal to or below 0.5. Otherwise, if the *d* is above 0.5 and below 1 the process is mean reverting; otherwise, innovation has no long-term impact on future values. The main research findings illustrated that of all G7 markets, FTSE 100 is the only market exhibiting evidence of the existence of a long-term memory process that could be explained by the disruption created by the BREXIT process. According to Breinlich et al.

(2018); Sezgin et al. (2021), Ben Ameur and Louhichi (2022); Qiao et al. (2021), BREXIT had a substantial negative effect on FTSE 100 volatility. The FTSE 100 was in the middle of sustained volatility, while other markets reacted positively. Moreover, our long memory process support patterns exhibited by the series descriptive statistics mean results, as the FTSE 100 has the lowest mean return among all G7 markets with 0.000103 percent. In the case of E7 markets, the FIGARCH (1,1) offered better results than G7 markets, as it captured long memory volatility for BIST 100, IDEX, BSE 100 markets that exhibited evidence of the existence of a long-term memory process. According to Su (2021), the E7 markets have higher returns and higher levels of risk than the G7 markets. This fact is reflected in the E7 volatility that ranged between 8 to 56 days, giving a gap of 48 days. Compared with the volatility lasting effects for G7 markets, which are more developed economies, the range was between 14 to 24 days, giving a gap range of 10 days. In the GCC markets, Bahrain is the only market exhibiting evidence of a long-term memory process, as Bahrain recorded FIGARCH (1,1) (0.03101) when the range of the GCC markets was between 0.60103 and 1.175244, which illustrates that Bahrain is acting differently from the rest of GCC markets. Moreover, for the GARCH (1,1), Bahrain had the lowest volatility persistence 0.796208, indicating a low volatility level compared to the GCC, ranging between 0.957184 and 0.972619. The volatility lasting effects for Bahrain is 4 days, and the range for the rest of the GCC was between 16 and 25 days. Our findings for Bahrain volatility persistence and long memory volatility aligned with the descriptive analysis results. Bahrain had the lowest standard deviation among all GCC countries 0.005143 percent, and the range was between 0.007999 and 0.010979. The results can be explained due to the fact that Bahrain is considered a very small economy compared to GCC countries.

In this study, we discovered that the standard deviation as volatility measurement and persistence does not match and sometimes has inverted results. For example, the G7 market SP-TSX is less volatile according to the standard deviation, while at the same time, it recorded the highest volatility persistence. According to Bentes and Cruz (2011) argument, smaller markets are characterised by less liquidity, and as such, they are less efficient in the sense of the Efficient Market Hypothesis as outlined in the seminal papers by Fama et al. (1969), therefore exhibiting higher persistence. This argument is supported with the findings of Di Matteo et al. (2003); Grau-Carles (2000) and also by this research paper findings.

*Research Limitations*

This research study offers interesting insights into the performance of the Kuwait stock market during times of significant uncertainty, as experienced during the 2020 dual shock (oil prices war and global health crisis). Although our study provides significant findings, it also has some limitations. The study is limited to the analysis of the impact of oil prices and major global markets on the Kuwait stock market. The study could be improved by integrating additional volatility models that help to examine volatility's lasting effects and spillover dynamics. Furthermore, the research study could consider exploring macroeconomic fundamentals, for example inflation rates, money supply, interest rates, unemployment rates and GDP performance, to examine to which extent they are linked to the dynamics of the studied stock markets.

## 6. Conclusions

Volatility is considered the most common measure of risk and is very helpful when assessing financial markets' uncertainty. Volatility modelling allows investors to capture potential losses and investment opportunities, monitor their investments, and consider the importance of hedging techniques and strategies to counter and manage the effects of market uncertainty. This study offers interesting insights into the dual impact of the global health crisis and the oil price shock that took place in 2020 and has a significant effect on the global economic and financial system, particularly in the case of Kuwait, a small oil-exporting economy. The study examined how the world's major markets,

the G7, E7, GCC and oil benchmarks WTI, Brent, Dubai and OPEC reacted during the dual shock. By implementing volatility models based on the well-known GARCH and FIGARCH models (widely recognised and applied in the academic literature and being considered as a core reference point in the stock markets evaluation), this paper tested the existence of volatility persistence and long memory processes. The core research findings showed that the FIGARCH model did not perform well except for the cases of FTSE 100 in G7 countries, BIST 100, IDEX, BSE 100 in E7 countries, and Bahrain in GCC countries. For the case of oil benchmarks, the FIGARCH model did not work at all, signifying and highlighting the importance of the GARCH model that emerges as the dominant model with robust outcomes. There were significant differences between the markets analysed, as Kuwait emerged as one of the stable markets within the period because of the lower drops experienced compared with the examined markets. Furthermore, the study's core research findings provide interesting insights to investors as the Kuwait stock market offers diversification opportunities and could act as a leveller during times of significant uncertainty. The Kuwait stock market can be viewed as an attractive destination for investors when they develop their portfolios because, in the context of G7, E7 and GCC economies, Kuwait is emerging as one of the most stable markets. This means that the Kuwait stock market could play a role in the design of long-term investment portfolios.

The empirical contribution of this study is threefold: (i) the existing literature has not provided evidence of market performance amidst the 2020 dual market shock within the context of volatility persistence and long-memory processes at the global level; (ii) the study provides critical and valuable insights for investment portfolio managers seeking to diversify their portfolio composition, for corporate financial decision-makers and investors seeking to hedge against market uncertainty derived from shocks that destabilise macroeconomic fundamentals, as the Kuwait stock market provides evidence of differing market reactions; (iii) finally, policymakers need to consider the importance of economic diversification for countries such as Kuwait that is over-reliant on oil and natural gas that account for nearly 60% of GDP and about 92% of export revenues. Economic diversification and sustainability are important aspects to consider as the country seeks to reexamine its dependency on fossil fuels.

## 7. Patents

This section is not mandatory but may be added if there are patents resulting from the work reported in this manuscript.

**Author Contributions:** Both authors made a contribution in every section of the paper. All authors have read and agreed to the published version of the manuscript.

**Funding:** This research received no external funding personal funding.

**Data Availability Statement:** Daily time series data are collected for G7, E7, GCC and four crude oil benchmarks represented by the US West Texas Intermediate (WTI), the European Brent Index, Dubai Crude oil (Dubai) and OPEC reference basket (OPEC). G7, E7, GCC and Oil benchmarks were chosen based on secondary data for continuous returns downloaded from DataStream. In case of request, I provide all the requested data.

**Acknowledgments:** We acknowledge the anonymous reviewers for their useful comments that enabled us to improve this manuscript.

**Conflicts of Interest:** The authors declare that they have no known competing financial interests or personal relationships that could have appeared to influence the work reported in this paper.

## Appendix A

**Table A1.** Research Studies Examining Oil Prices Fluctuations in the Case of Kuwait.

| Authors | Country | Variables | Period | Outcomes |
|---|---|---|---|---|
| **Al-Shami and Ibrahim (2013)** | Kuwait | inflation rate, money supply, interest rate, oil prices and unemployment rate. | January 2001 to December 2010 | A positive relation between inflation rate, money supply (M2), oil prices, and stock returns. |
| **Al Hayky and Naim (2016)** | Bahrain, Qatar, Kuwait, Saudi Arabia, U.A.E. Oman | Dynamic relationship between oil prices and Kuwait stock exchange index | monthly-data from November 2006 to February 2015 | The Kuwait stock market index has a positive and significant relationship with oil prices during the high volatility periods but no relationship during low volatility periods |
| **Merza and Almusawi (2016)** | Kuwait | inflation rate, money supply, interest rate, oil prices and unemployment rate. | January 2001 to December 2010 | A positive relationship between the inflation rate, money supply (M2) and oil prices and stock returns. |
| **Kisswani and Elian (2017)** | Kuwait | Relationship between Kuwait stock market (KSE) and oil prices (Brent and WTI) at a sectoral level The study analysed ten major sectors in Kuwait | 3 January 2000, until 9 December 2015, for | The systematic long-run effect between oil prices and some Kuwait sectoral stock prices. The empirical result offers evidence of a short-run systematic effect in the case of WTI price, but no evidence of a systematic effect was found in the case of Brent. |
| **Elian and Kisswani (2018)** | Kuwait | Oil prices (Brent) (WTI) affect the stock market returns in the context of 'Kuwait stock market' (KSE.) | daily data 3 January 2000 until 9 December 2015 | There is a long-run relationship between Kuwait stock market returns and both oil prices (Brent and WTI) in which the daily oil price shocks have a negative impact on stock returns. |
| **Al-Kandari and Abul (2019)** | Kuwait | M2, three months deposit interest rate, oil prices, US to Kuwaiti dinar exchange rate and inflation rate. | monthly data 2005–2018 | The study confirms a short-run relationship between oil and Kuwait stock market. |
| **Alshihab and Al Shammari (2020)** | Kuwait | Fluctuations of the oil prices on the Kuwait stock market returns | month-to-month period of 2000 to 2020 | The long run showed that the price of oil has a positive relationship with stock market returns. The study confirmed that changes in Kuwaiti stock market returns are affected by oil price fluctuations in the short run. |
| **Al Refai et al. (2022)** | Bahrain, Qatar, Kuwait, Saudi Arabia, U.A.E. Oman | examined the impact of Covid-19 cases and oil prices shocks | sub-sample 5 January 2017, to 10 March 2020, and 11 March to 17 September 2020 | Their findings illustrate that Kuwait's stock market responded to positive and negative oil price shocks. |

Source: Authors (2022).

## Appendix B

**Table A2.** Data Set.

| G7 Countries | | |
|---|---|---|
| **Country** | **Short-Form** | **Definition** |
| **Germany** | DAX 30 | Deutsche Altien Xchange (DAX) Performance Index |
| **U.S.A** | Dow Jones Industrials | The Dow Jones Industrial Average |
| **France** | CAC-40 | Cotation Assistee en Continu (CAC) 40 Index |

**Table A2.** *Cont.*

| G7 Countries | | |
|---|---|---|
| **UK** | FTSE 100 | Financial Times stock exchange FTSE100 index |
| **Italy** | FTSE-MIB | Milano Indice di Borsa |
| **Japan** | NIKKEI 225 | NIKKEI 225 index |
| **Canada** | S&P-TSX | S&P-TSX composed index |
| **USA** | S&P 500 | Standard and Poor 500 composite index |
| **E7 Countries** | | |
| **BRAZIL** | BOVESPA | Brasil Bolsa Balcão |
| **TURKEY** | BIST NATIONAL 100 | Borsa İstanbul |
| **RUSSIA** | RTS INDEX | 50 Russian stocks traded on the Moscow Exchange |
| **MEXICO** | IPC (BOLSA) | Bolsa Mexica de Valores |
| **INDONEASIA** | IDEX | Indonesia Stock Exchange (IDX) |
| **CHINA** | SHANGHAI | Shanghai Composite Index |
| **INDIA** | BSE | Bombay Stock Exchange (BSE.) |
| **GCC countries** | | |
| **UAE** | DUBAI | Dubai Financial Market |
| **Qatar** | Qatar | Qatar Composite Index |
| **SAUDI** | TASI | Saudi Stock Exchange |
| **Bahrain** | Bahrain | Bahrain All Share Index |
| **Kuwait** | BKA. | Boursa Kuwait |
| **Oman** | Oman | Muscat Security Market |
| **Oil Prices** | | |
| **USA** | WTI | West Texas |
| **U.K** | Brent | Brent Blend |
| **UAE.** | Dubai | Dubai |
| **OPEC** | OPEC | OPEC Reference basket |

## Notes

[1] See the following link from the WHO at: https://covid19.who.int/table, accessed on 11 March 2020.

[2] For the sake of brevity the results from the residual checks are not included in the paper, but they are available upon request.

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
