# Peer review of "Financial Uncertainty from a Dual Shock at Global Level–Insights from Kuwait"

_ijfs, doi:10.3390/ijfs10040101_

Round 1

Reviewer 1 Report

Dear Authors,

1. The study is Kuwait as you mentioned in the title but if you read the abstract and Introduction, you will feel the study is on  G7, E7 and the Gulf region stock markets.

2. Please adjust the abstract to include clearly the objective of your study, methods, main results and policy implications.

3. Please adjust the introduction to include the motivation, gap, objectives, and the structure of the following section. Also, please enhance the flow of information especially for Kuwait. 

4. Please have one introduction section  that includes 1.1 2020 Dual Economic Shock . then  have a seperate section for Literature Review  that includes latest studies from Kuwait, GCC, G7, E7 in order to help you to link your results in the discussion part.

5. For Oil Prices, which price you take? For UAE, you put Duabi. What does it mean? You can consider WTI for oil prices as a unified index as mentioned in the literature. 

6.  Where is the robustness of your results, in checking return volatility we have many models, so please use more models to check the robustness and reliability of your results.

7. Please add more policy implications, limitations and scope fir future work

8. Please do proofreading for the whole manuscript.

Author Response

Response to received comment

Dear reviewer 1

Thank you for your useful comments that enabled us to improve the manuscript.

All comments were addressed and highlighted the changes in Yellow in the document.

  1. The study is Kuwait as you mentioned in the title but if you read the abstract and Introduction, you will feel the study is on  G7, E7 and the Gulf region stock markets.

Title of the paper changed to (Financial Uncertainty from a Dual Shock on Global Major Markets – Insights from Kuwait)

  1. Please adjust the abstract to include clearly the objective of your study, methods, main results and policy implications.

The abstract is adjusted with objectives, methods and results as follows:

Consequently, this research paper examines how the dual shock impacted the Kuwait stock exchange and some of the world's most prominent economies represented by the G7, E7 and the Gulf region. The analysis seeks to better understand the dynamics of financial linkages and their potential spillover effects at a global scale. The study examines the impact of the global health crisis and the oil price shock and their dual impact on the GCC economies due to the critical implications for the Gulf region, which is heavily reliant on oil. Volatility modelling was considered, and the GARCH and FIGARCH modelling were selected as they are widely studied and recognised in the academic literature. Daily returns for the period December 31 2015 to December 9 2021 where considered for the volatility analysis. The research findings suggest that the GARCH(1,1) captured volatility persistence dynamics in the studied markets. While the FIGARCH (1,1) did not offer significant evidence of long memory processes affecting the studied markets except for the cases of FTSE 100, BIST 100, IDEX, BSE 100 and Bahrain.

  1. Please adjust the introduction to include the motivation, gap, objectives, and the structure of the following section. Also, please enhance the flow of information especially for Kuwait

The introduction is adjusted with motivation and objectives and paper structure as follows:

Undoubtedly, oil plays a critical role in Kuwait's economy as the country has a significant time-varying financial dependency on fossil fuels. Oil and natural gas account for nearly 60% of GDP and about 92% of export revenues. Within the outlined context, the purpose of this paper is to analyse the impact of COVID-19 and the oil shock on the return and volatility of the Kuwait stock exchange along with eight major markets indices of the top G7 (the world most developed economies), E7 (Emerging Economies) and the GCC countries. The reviewed literature reveals that existing research studies have not integrated the world's major stock markets to frame the performance of the Kuwait stock exchange (Boursa) amidst a dual market shock named the global health crisis and the oil price war in 2020. Furthermore, this research study is supported by two GARCH specifications, namely GARCH and FIGARCH models seeking to examine volatility persistence and long memory processes. Volatility modelling is implemented to explore the performance of the Kuwaiti stock exchange among the world's major indices and to examine volatility patterns to identify which markets have shown more resilience to the 2020 dual shock. The results showed that the GARCH(1,1) helped to explain volatility persistence dynamics in the studied markets. However, the FIGARCH (1,1) did not offer significant evidence of long memory processes affecting the studied markets except for FTSE 100, BIST 100, IDEX, BSE 100, and Bahrain. Section 2 presents a review of the relevant literature; section 3 defines the methodological research framework. The paper's main results and discussion are presented in sections 4 and 5, and finally, section 6 concludes the paper.

  1. Please have one introduction section  that includes 1.1 2020 Dual Economic Shock . then  have a seperate section for Literature Review  that includes latest studies from Kuwait, GCC, G7, E7 in order to help you to link your results in the discussion part.

The structure of the paper has been updated and  the introduction and literature review are presented in different sections as requested by the reviewer please see page 2 and 3 in the updated manuscript.

  1. For Oil Prices, which price you take? For UAE, you put Duabi. What does it mean? As mentioned in the literature, you can consider WTI for oil prices as a unified index. 

The WTI for is a unified oil benchmark but we also included 3 crude oil benchmarks (WTI, Brent, Dubai and OPEC) were tested. The rational was to offer a broader context to existing dynamics.

  1. Please add more policy implications, limitations and scope fir future work

a new section was added in the paper as research limitation:

Research Limitations

This research study offers interesting insights into the performance of the Kuwait stock market during times of significant uncertainty, as experienced during the 2020 dual shock (oil prices war and global health crisis). Although our study provides significant findings, it also has some limitations. The study is limited to the analysis of the impact of oil prices and major global markets on the Kuwait stock market. The study could be improved by integrating additional volatility models that help to examine volatility's lasting effects and spillover dynamics. Furthermore, the research study could consider exploring macroeconomic fundamentals for example: inflation rates, money supply, interest rates, unemployment rates and GDP performance to examine to which extent they are linked to the dynamics of the studied stock markets. Hence, policy implications is added in the introduction, (finally, policymakers need to consider the importance of economic diversification for countries like Kuwait that is overreliant on oil and natural gas that account for nearly 60% of GDP and about 92% of export revenues. Economic diversification and sustainability are important aspects to consider as the country seeks to reexamine its dependency on fossil fuels.)

Please let us know if you have any suggestions.

Best regards

Talal Alotaibi

[email protected]

Reviewer 2 Report

In the introduction there can be put in the summary some of the findings of the research, beside its objectives.

Author Response

Response to received comment

Dear reviewer 2

Thank you for your useful comments that enabled us to improve the manuscript.

Concerning your comment, “In the introduction there can be put in the summary some of the findings of the research, beside its objectives”. I would like to inform you that the updated draft has integrated your suggestions as part of the introduction where we have now included a summary of the research findings and the paper objectives as requested.

Undoubtedly, oil plays a critical role in Kuwait's economy as the country has a significant time-varying financial dependency on fossil fuels. Oil and natural gas account for nearly 60% of GDP and about 92% of export revenues. Within the outlined context, the purpose of this paper is to analyse the impact of COVID-19 and the oil shock on the return and volatility of the Kuwait stock exchange along with eight major markets indices of the top G7 (the world most developed economies), E7 (Emerging Economies) and the GCC countries. The reviewed literature reveals that existing research studies have not integrated the world's major stock markets to frame the performance of the Kuwait stock exchange (Boursa) amidst a dual market shock named the global health crisis and the oil price war in 2020. Furthermore, this research study is supported by two GARCH specifications, namely GARCH and FIGARCH models seeking to examine volatility persistence and long memory processes. Volatility modelling is implemented to explore the performance of the Kuwaiti stock exchange among the world's major indices and to examine volatility patterns to identify which markets have shown more resilience to the 2020 dual shock. The results showed that the GARCH(1,1) helped to explain volatility persistence dynamics in the studied markets. However, the FIGARCH (1,1) did not offer significant evidence of long memory processes affecting the studied markets except for FTSE 100, BIST 100, IDEX, BSE 100, and Bahrain. Section 2 presents a review of the relevant literature; section 3 defines the methodological research framework. The paper's main results and discussion are presented in sections 4 and 5, and finally, section 6 concludes the paper.

Please check the updated draft and let us know if you have any suggestions.

Best regards

Talal Alotaibi

[email protected]

Reviewer 3 Report

The paper addresses interesting and important topic of the effects of the financial shocks observed since 2020 on the financial markets, using the example of the stock market in Kuwait as well as in some other emerging and advanced economies. The study is based on sound methodology and the results are interpreted and discussed correctly. There are some rather minor issues that could be changed in order to improve the paper even further.

1. The aims of the study presented throughout the paper are inconsistent and, even more importantly, some of them do not fully reflect the discussed issues. For instance, the title of the paper refers exclusively to Kuwait whereas the aim presented in the abstract mentiones more economies.

2. The structure of the paper could be improved. For example, detailed literature review could be presented separately rather than being a part of rather long introductory section.

3. The last Section requires some extensions. Limitations of the analysis as well as directions for the future studies should be provided.

4. I recommend using a larger set of references - some more general topics covered are well discussed in many publications.

Author Response

Response to received comment

Dear reviewer 3

Thank you for your useful comments that enabled us to improve the manuscript.

All comments from 1-4 were addressed and highlighted the changes in Yellow in the document.

1.The aims of the study presented throughout the paper are inconsistent and, even more importantly, some of them do not fully reflect the discussed issues. For instance, the title of the paper refers exclusively to Kuwait whereas the aim presented in the abstract mentions more economies.

The abstract was updated to fulfil the mentioned comments.

Abstract: The emergence of the COVID-19 pandemic has caused financial markets to suffer historic losses during the first quarter of 2020 at levels unseen since the crises of the futures markets in 1987. In January 2020, the World Health Organization officially declared the pandemic, bringing significant uncertainty levels to the global economy. The COVID-19 pandemic affected global markets, parallel to a simultaneous shock derived from the Saudi Arabia and Russia oil price war. As the worldwide health crisis escalated, governments took different measures that led to economic lockdowns that caused significant disruption of global supply chains and reduced aggregated demand during 2020. These circumstances fueled oil market volatility with lasting effects as the global economy now faces high inflationary pressures derived from heightened energy costs.

Consequently, this research paper examines how the dual shock impacted the Kuwait stock exchange and some of the world's most prominent economies represented by the G7, E7 and the Gulf region. The analysis seeks to better understand the dynamics of financial linkages and their potential spillover effects at a global scale. The study examines the impact of the global health crisis and the oil price shock and their dual impact on the GCC economies due to the critical implications for the Gulf region, which is heavily reliant on oil. Volatility modelling was considered, and the GARCH and FIGARCH modelling were selected as they are widely studied and recognised in the academic literature. Daily returns for the period December 31 2015 to December 9 2021 where considered for the volatility analysis. The research findings suggest that the GARCH(1,1) captured volatility persistence dynamics in the studied markets. While the FIGARCH (1,1) did not offer significant evidence of long memory processes affecting the studied markets except for the cases of FTSE 100, BIST 100, IDEX, BSE 100 and Bahrain.

  1. The structure of the paper could be improved. For example, detailed literature review could be presented separately rather than being a part of rather long introductory section.

The structure of the paper is updated, the introduction and literature review are now presented in different sections. Please check the paper page 2 and 3.

  1. The last Section requires some extensions. Limitations of the analysis as well as directions for the future studies should be provided.

Thank you for your comment and suggestions. The paper has been updated and a new section was added to include the core research limitations.

Research Limitations

This research study offers interesting insights into the performance of the Kuwait stock market during times of significant uncertainty, as experienced during the 2020 dual shock (oil prices war and global health crisis). Although our study provides significant findings, it also has some limitations. The study is limited to the analysis of the impact of oil prices and major global markets on the Kuwait stock market. The study could be improved by integrating additional volatility models that help to examine volatility's lasting effects and spillover dynamics. Furthermore, the research study could consider exploring macroeconomic fundamentals for example: inflation rates, money supply, interest rates, unemployment rates and GDP performance to examine to which extent they are linked to the dynamics of the studied stock markets.

  1. I recommend using a larger set of references - some more general topics covered are well discussed in many publications.

I careful review of the literature was conducted and additional research was included to support the paper. Three additional references were added

  • Al-Refai, H., Zeitun, R., & Eissa, M. A. A. (2022). Impact of Global Health Crisis and Oil Price Shocks on Stock Markets in the GCC. Finance Research Letters, 45, Article ID: 102130.
  • Abdulrazaq, Y. M., & Shetty, S. (2020). Oil Sector Spillover Effects to the Kuwait Stock Market under Uncertainty. International Journal of Accounting & Finance Review, 5,32-41.
  • Al-Kandari, A. M., & Abul, S. J. (2019). The impact of macroeconomic variables on stock prices in Kuwait. International Journal of Business and Management14(6), 99-112.

Please let us know if you have any suggestions.

Best regards

Talal Alotaibi

[email protected]

Round 2

Reviewer 1 Report

The authors addressed my points. Accept

Author Response

Dear reviewer

Thank you

Regards

Talal